# LRP1 Deficiency Promotes Mitostasis in Response to Oxidative Stress: Implications for Mitochondrial Targeting after Traumatic Brain Injury

**DOI:** 10.3390/cells12101445

**Published:** 2023-05-22

**Authors:** Gopal V. Velmurugan, W. Brad Hubbard, Paresh Prajapati, Hemendra J. Vekaria, Samir P. Patel, Alexander G. Rabchevsky, Patrick G. Sullivan

**Affiliations:** 1Spinal Cord and Brain Injury Research Center, University of Kentucky, Lexington, KY 405036, USA; vetvelu@uky.edu (G.V.V.); bradhubbard@uky.edu (W.B.H.); paresh.prajapati@uky.edu (P.P.); hemendravekaria@uky.edu (H.J.V.); samir.patel@uky.edu (S.P.P.); agrab12@uky.edu (A.G.R.); 2Department of Neuroscience, University of Kentucky, Lexington, KY 40536, USA; 3Lexington Veterans’ Affairs Healthcare System, Lexington, KY 40502, USA; 4Department of Physiology, University of Kentucky, Lexington, KY 40536, USA

**Keywords:** Traumatic Brain Injury (TBI), LDL receptor-related protein 1 (LRP1), mitochondria, mitostasis, oxidative stress, mitochondrial morphology, mitochondrial network

## Abstract

The brain undergoes oxidative stress and mitochondrial dysfunction following physiological insults such as Traumatic brain injury (TBI), ischemia-reperfusion, and stroke. Pharmacotherapeutics targeting mitochondria (mitoceuticals) against oxidative stress include antioxidants, mild uncouplers, and enhancers of mitochondrial biogenesis, which have been shown to improve pathophysiological outcomes after TBI. However, to date, there is no effective treatment for TBI. Studies have suggested that the deletion of LDL receptor-related protein 1 (LRP1) in adult neurons or glial cells could be beneficial and promote neuronal health. In this study, we used WT and LRP1 knockout (LKO) mouse embryonic fibroblast cells to examine mitochondrial outcomes following exogenous oxidative stress. Furthermore, we developed a novel technique to measure mitochondrial morphometric dynamics using transgenic mitochondrial reporter mice mtD2g (mitochondrial-specific Dendra2 green) in a TBI model. We found that oxidative stress increased the quantity of fragmented and spherical-shaped mitochondria in the injury core of the ipsilateral cortex following TBI, whereas rod-like elongated mitochondria were seen in the corresponding contralateral cortex. Critically, LRP1 deficiency significantly decreased mitochondrial fragmentation, preserving mitochondrial function and cell growth following exogenous oxidative stress. Collectively, our results show that targeting LRP1 to improve mitochondrial function is a potential pharmacotherapeutic strategy against oxidative damage in TBI and other neurodegenerative diseases.

## 1. Introduction

Traumatic brain injury (TBI) causes neuronal damage followed by neurodegeneration and cognitive deficit [1,2]. According to the Center for Disease Control and Prevention (CDC), over 220,000 TBI-related hospitalizations and 64,000 deaths occurred in 2020 [3]. However, there is no effective Food-and-Drug-Administration (FDA)-approved drug for the treatment of TBI. At this juncture, it is important to understand the mechanism of the pathobiology of TBI in order to develop effective therapeutic targets. Mitochondrial reactive oxygen species (ROS) and changes in mitochondrial homeostasis have been reported in several metabolic, inflammatory, and neurodegenerative diseases [4,5,6,7]. Compelling experimental data from our group and others demonstrate that oxidative stress and mitochondrial dysfunction are linked to the neuropathological sequelae of TBI. Though there are studies focused on understanding mitochondrial dysfunction in TBI and other neurodegenerative diseases, there is no effective pharmacological treatment for TBI which warrants further studies. Hence, identifying molecular targets which may protect mitochondria, improve mitochondrial biogenesis/bioenergetics, and decrease oxidative stress is of great importance to combat TBI and neurodegenerative diseases. 

Low-density lipoprotein receptor-related protein (LRP1) is a large (600 kDa) endocytic molecule that belongs to the LDL receptor gene family, expressed ubiquitously in various cell types [8]. LRP1 regulates signaling pathways by shedding the extracellular soluble domain (sLRP1) and cleaving the intracellular domain (LRP1-ICD) [9,10]. In macrophages, LRP1 is known to regulate inflammation and cholesterol accumulation [11]. In the brain capillary endothelial cells, LRP1 facilitates Aβ efflux from the brain side to the blood side [12]. In the liver, LRP1 works in a manner similar to LDL receptors in the plasma clearance of apo-E-containing lipoproteins [13]. In smooth muscle cells and fibroblasts, LRP1 deficiency generates a proliferative phenotype by increasing platelet-derived growth factor receptor-β (PDGFR-β) and transforming growth factor-β (TGF-β) expression [14]. In neurons and astrocytes, LRP1 plays a significant role in spreading pathological α-synuclein, tau, and amyloid-β (Aβ) [15,16]. Though LRP1 plays critical roles in endocytosis and cell signaling, LRP1 deletion is known to be beneficial in some pathophysiological aspects of the brain. For instance, conditional LRP1 knockout (LKO) in oligodendrocyte progenitor cells (OPCs) from adult mouse brains increases proliferation and the number of newborn mature oligodendrocytes [17]. LRP1 gene silencing or the pharmacological inhibition of LRP1 increases neurite outgrowth [18]. The deletion of LRP1 in microglia blocks neuro-inflammation and neuropathic pain processing [10]. In vascular endothelial cells, LRP1 deletion improves glucose homeostasis in the brain [19] and increases proliferation and angiogenesis through Poly ADP ribose (PARP-1) [20], which, in turn, regulates nuclear factor erythroid 2 (NFE2)-related factor 2 (NRF2), the master regulator of antioxidants [21]. Although evidence suggests that LRP1 plays important roles in cell proliferation, migration, and energy homeostasis [22] and responds to cellular stress conditions such as oxidative stress, hypoxia, tissue injury, and inflammation [23,24,25], the way in which LRP1 regulates mitochondrial function and ROS is not known. Brain mitostasis and mitochondrial energy metabolism are recognized as critical regulators of neuronal health and survival [26]. Acute and chronic oxidative stress negatively regulates mitochondrial dynamics and function, which become detrimental to cell survival. Our evidence demonstrates that LRP1 deficiency protects mitochondrial function and mitostasis [27] from oxidative stress by stimulating pathways related to mitochondrial biogenesis and antioxidants. 

## 2. Methods

### 2.1. Animals

All the studies performed were approved by the University of Kentucky IACUC, which is accredited by the Association for the Assessment and Accreditation for Laboratory Animal Care, International (AAALAC, International), and all experiments were performed in accordance with its guidelines. All animal experiments were compliant with the ARRIVE guidelines, and the experiments were carried out in accordance with the National Institutes of Health guide for the Care and Use of Laboratory Animals (NIH Publications No. 8023, revised 2011). Mice (B6;129S-Gt (ROSA)26Sortm1.1 (CAG-COX8A/Dendra2) Dcc/J) expressing a mitochondrial-specific version of Dendra2 green (mtD2g; heterozygous) monomeric fluorescent protein were cryo-recovered from Jackson Laboratories (Strain#: 018397) and transported to the Division of Laboratory Animal Resources (DLAR) at the University of Kentucky. After acclimatization, the mice were bred and maintained as a homozygous colony. All experiments were performed using homozygous D2g mice at 12–16 weeks of age with an average body weight of 25 g.

### 2.2. Controlled Cortical Impact

Controlled cortical impact (CCI) was performed as previously described (1.0 mm depth of contusion at 3.5 m/s with a dwell time of 500 ms) [28]. In brief, the animals were anesthetized using isoflurane (2–5%), shaved, cleaned, and prepared for the surgical procedure. A Kopf stereotaxic frame was used to generate brain injury under a pneumatic impactor (Precision Science Instruments). A longitudinal skin incision was made down the middle of the head dorsum before craniotomy (4 mm) was performed on the left side of the skull between the lambda and bregma. The mice were injured by hitting the dura of the brain using a 3 mm flat-tip impactor. The injury area was cleaned using cotton tips to mitigate any bleeding, the craniotomy was covered with SurgiSeal, and the wound was closed using staples. 

### 2.3. Cell Culture

Wild-type (WT) mouse embryonic fibroblast (MEF1; ATCC CRL-2214) and LRP1 knockout (LKO) MEF cells (ATTCC CRL-2216) from ATCC were generously provided by Dr. Florin Despa, University of Kentucky. The cells were maintained as recommended by the ATCC using DMEM media with 10% FBS. 2,2′-Azobis-2-methyl-propanimidamide dihydrochloride (AAPH), a free radical generator, was used to induce oxidative stress by lipid peroxidation in order to induce mitochondrial dysfunction [29]. The dose–response to AAPH (1, 2, 3, and 5 mM) and mitochondrial respiration (Appendix A) were assessed using a Seahorse XFe96 Analyzer, as described in the Methods sections. All the experiments were performed using the optimal concentration (1 mM) of AAPH at which we found no effect on mitochondrial respiration in the LKO cells and decreased mitochondrial respiration in the WT cells. The cells were treated overnight (16–18 h) for all experiments. 

### 2.4. Mito-Stress Test

Using a Seahorse XFe96 Flux Analyzer (Agilent Technologies, Palo Alto, CA, USA), a mitochondrial stress test was performed according to the manufacturer’s instructions. Briefly, WT and LKO cells were seeded at a density of 1 × 10^4^ cells per well on XF microplates. After 24 h, the cells were treated with either AAPH or a vehicle overnight. Then, the cells were incubated with XF assay medium together with substrates (10 mM glucose, 1 mM pyruvate, and 2 mM L-glutamine) for an hour before the oxygen consumption rate (OCR) measurement. After three measurements of the baseline OCR, we added respiratory chain inhibitors/uncoupler sequentially into each well, as follows: 1 µM Oligo, 4 µM FCCP, and 0.5 µM Rot/Ant. After each injection, an additional three OCR readings were taken. Different OCR parameters were calculated using Wave software version 2.6.1 (Agilent Technologies, Santa Clara, CA, USA) or manually. The final OCR measurements were normalized to the cell numbers. The experiment was repeated 5 times with at least 3 technical replicates each time. 

### 2.5. Mitochondrial Quantification in Imaris

Coronal brain sections (5 µm thick paraffin sections) taken 24 h post-CCI were dewaxed, rehydrated, and mounted on glass slides using Nucblue (P36981, Invitrogen, Waltham, MA, USA). Fluorescence Z-stack images were taken using a confocal microscope (Nikon A1R). The mitochondrial shape, size, and count were quantified as described previously [30]. Briefly, the Z-stack images were opened in Imaris (X64 9.6.1), and the surface creation tool was assigned to the green channel (mtD2g). The smoothing surface was enabled with the grain size of 0.5 µm to generate a cleaner border. Background subtraction (local contrast) was used to separate the mitochondria from the background. A consistent threshold was applied to all images during background subtraction. In some cases, manual adjustments were made, as needed, to the surface shape to reduce noise rather than changing the actual 3D model of the mitochondria. Split-touching object was enabled to separate individual adjacent mitochondria. The area and volume of individual mitochondria detected using Imaris were exported to an excel file. Different mitochondrial volumes (µm^3^) were frequently distributed with 1 bin width and 0.1–15 bin range (µm^3^). The percent distribution of mitochondrial volume was calculated by dividing the total number of mitochondria multiplied by 100 for each bin width. The sphericity of the mitochondria was directly calculated using Imaris software.

### 2.6. MitoTracker Imaging and Mitochondrial Network Analysis (MiNA)

WT and LKO cells were cultured in 24-well glass-bottomed plates and stained with prewarmed media containing 500 nM of MitoTracker Green TM (M7514, Thermofisher, Waltham, MA, USA) for 30 min at 37 °C. After washing twice, the cells were imaged under a confocal microscope (100X oil objective; Nikon with NIS-Elements version 5.30.05). At least 3–6 Z-stack confocal images were taken randomly in different fields in three different coverslips. Each microscopic field had at least 10 cells (total of 30–60 cells/group). Subsequently, the images were imported into ImageJ and analyzed with MiNA macro (developed by StuartLab) to measure the mitochondrial network parameters. The experiments were repeated twice to confirm consistent results. For the mtD2g mice (*n* = 3/group), optically zoomed confocal Z-stack images (2 images/side/animal were taken and averaged per animal) were used to quantify the mitochondrial network.

### 2.7. ROS Assay in Cells

WT and LKO cells were cultured in 24-well glass-bottomed plates. ROS production in the cells was determined using a highly sensitive DCFH-DA-ROS assay kit (Dojindo Molecular Technologies, Kumamoto, Japan) according to the manufacturer’s instructions. Fluorescence signals were observed using a fluorescence microscope (Nikon Ti2). Then, 10 cells were randomly selected from each field (8–10 microscopic fields/group) from three independent coverslips, and the mean fluorescence intensity (green channel) was measured after subtracting the background fluorescence using NIS-Elements version 5.30.05 software.

### 2.8. Cell Proliferation Assay

WT and LKO cells were cultured in 24-well glass-bottomed plates and treated with a vehicle or APPH. At the end of the treatment, the cells were incubated with EdU (5–ethynyl–2′–deoxyuridine), dissolved in DMSO (final concentration of 10 uM), and fixed with 10% buffered formalin for 15 min. The incorporated EdU was detected using a Click-iT EdU Alex fluor 488 imaging kit according to manufacturers’ instructions (C10632, Thermo fisher, Waltham, MA, USA). Glass wells were covered using a Prolong Glass antifade mount with Nucblue (P36981, Invitrogen, Waltham, MA, USA) and imaged (6 random fields/group from three different coverslips; each field had more than 200 cells) using a fluorescent microscope (Nikon Ti2). The number of cells in each field was counted using ImageJ software. The DAPI (blue) nuclear represented the total number of cells, and Edu (green) nuclear represented proliferating cells. The proliferation rate was calculated as: (EdU^+^ nuclear number/DAPI^+^ nuclear number) × 100%.

### 2.9. Western Blot

Western blot analysis was performed for 4-HNE, LRP1, mitochondrial OXPHOS complex proteins, NDUFS1, PDGFR-β, and TFAM. Cell lysates were formed using RIPA buffer (150 mM NaCl, 1% Triton X-100, 0.5% sodium deoxycholate, 0.1% SDS, 50 mM Tris, pH 8.0) and centrifuged at 16,100× *g* for 30 min, and the total protein levels were estimated from the supernatant using a BCA kit (23225, Thermofisher). Western blot samples were obtained using XT sample buffer (1610791, Biorad, Hercules, CA, USA) with DTT and boiled at 95 °C for 10 min. The samples (15–20 µg protein, Eastbourne, UK) were resolved in duplicate 4–12% BIS-TRIS gel (3450125, Biorad, Hercules, CA, USA) under reducing conditions and transferred to the PVDF membrane. Probing was performed against 4-HNE (1:2500; HNE11-S, alpha diagnostic international, San Antonio, TX, USA), LRP1 (1:1000; sc-57351, Santa Cruze biotechnology, Dallas, TX, USA), NDUFS1 (1:1000, ab169540, abcam, Cambridge, UK), total OXPHOS antibody cocktail (1:1000; ab110413, abcam), mtTFA (1:1000, ab131607, abcam), and beta-actin (1:5000; 8H10D10, Cell Signaling, Danvers, MA, USA). The signals were detected using IRDy 68RD goat anti-mouse (1: 10,000; 926-68070, Li-Cor, Lincoln, NE, USA) and IRDye 800 CW goat anti-rabbit (1:10,000; 926-32211, Li-Cor). The protein levels were quantified via densitometric analysis using ImageJ software.

### 2.10. Immunofluorescence

Brain samples were collected 24 h post-CCI from the mtD2g mice and were formalin-fixed and paraffin-embedded. Coronal brain sections were antigen-retrieved in citrate buffer, permeabilized, and blocked together in 0.2% Trion X-100 in TBST + 1%BSA and 10% normal horse serum for one hour at room temperature (RT). Then, the sections were incubated overnight at 4 °C with primary antibody 4-HNE antibody (1:250; HNE11-S, alpha diagnostic international, San Antonio, TX, USA) in 50% diluted blocking buffer. Alexa flour 594 donkey anti-rabbit (1:500; A212207, Invitrogen) was used as a secondary antibody at RT for 1 h. After washing, the samples were mounted on glass slides using a Prolong Glass antifade mount with Nucblue (P36981, Invitrogen, Waltham, MA, USA). Images were acquired using a Nikon widefield fluorescence microscope with NIS-Elements version 5.30.05. The 4-HNE mean intensity was measured using ImageJ software (*n* = 3/group). 

### 2.11. Mitochondrial DNA Copy Number (mtDNA-CN) Estimation

mtDNA-CN was estimated as described previously [29], with some modifications. Total was DNA isolated from the vehicle- and AAPH-treated WT and LKO cells using a Qiagen DNA isolation kit. The mitochondrial subunit ND1 (mtND1) gene encoded by the mitochondrial genome was used as a maker for measuring the mitochondrial DNA content. mtDNA-CN was measured via qPCR using the following primer pair: ND1 Sense: 5′-TGAATCCGAGCATCCTACC-3′; and ND1 Antisense: 5′-ATTCCTGCTAGGAAAATTGG-3′. Then, it was normalized to the nuclear-encoded gene β-actin using the following primer pair: Actin Sense: 5′-GGGATGTTTGCTCCAACCAA-3′; and Actin Antisense: 5′-GCGCTTTTGACTCAGGATTTAA-3′.

### 2.12. qRT-PCR

Total RNA was isolated from the vehicle- and AAPH-treated WT and LKO cells using the TRIzol method. Quantitative real-time PCR (qRT–PCR) was performed using Taqman probes against genes (list of gene panel, Table 1) involved in mitochondrial biogenesis and antioxidants in a 384-well plate. The relative quantitative evaluation of gene levels was performed using the 2^−ΔΔCt^ method with 18 s RNA as an internal reference.

### 2.13. Statistics

Statistical analysis was performed using Graph Pad Prism 9 (GraphPad Software, San Diego, CA, USA). For all analyses, a significant difference between groups was defined as *p* < 0.05. Data are reported as the mean ± SD. The results were compared using a two-tailed Student’s t-test for two groups and two-way ANOVA followed by Sidak’s post hoc multiple comparison test, where appropriate, for multiple groups.

## 3. Results

### 3.1. Traumatic Brain Injury Induces Oxidative Stress and Mitochondrial Fragmentation in mtD2g Mice

To determine changes in mitostasis following TBI, we used a novel technique to measure mitochondrial dynamics in mtD2g mice at 24 h post-TBI. Representative micrographs (Figure 1A,B) of the mtD2g mice brain sections showed decreased mitochondrial fluorescence (density) in and around the injured area, compared to the contralateral side, indicative of mitochondrial damage. From the confocal Z-stack images, the mitochondrial number, shape, and size (in volume) were calculated using Imaris software. The representative volume–area distribution graph generated using Imaris (Figure 1C) and the percentage of mitochondrial volume distribution (Figure 1D) on the ipsilateral side demonstrated a leftward shift compared to the contralateral side, which suggested an increase in mitochondrial fission around the core of the injured cortex. Consistently, the shape of the mitochondria was significantly more spherical on the ipsilateral side compared to the long, elongated shape on the contralateral side (Figure 1E,F). Increased mitochondrial fission and a decreased mitochondrial branch length are known to be an ongoing pathological hallmarks of oxidative stress [31,32]. Accordingly, using the mitochondrial network analysis (MiNA) macro with ImageJ, it was found that the mito-footprint (Figure 1G), mean branch length (Figure 1H), mean summed branch length (Figure 1I), and mean network branch length (Figure 1J) were significantly decreased on the ipsilateral side compared to the contralateral side. Moreover, in the same brain samples, oxidative-stress-mediated lipid peroxidation quantified using 4-HNE immunostaining (Figure 1K,L) was found to be significantly increased on the ipsilateral compared to the contralateral side.

### 3.2. LRP1 Deficiency Improves Mitochondrial Bioenergetics following Oxidative Stress

To assess the role of LRP1 in oxidative stress, we used WT and LKO cells treated with AAPH (1 mM) and measured the mitochondrial function using a Seahorse XFe96 Analyzer. The representative Western blot results from the WT and LKO cells show the efficiency of LRP1 knockout in the LKO cells (Figure 2A). AAPH induces mitochondrial dysfunction through free radical generation that induces lipid peroxidation and mitochondrial swelling [29]. Consistently, 4-HNE adducts were increased dose-dependently in the whole-cell lysate from the WT cells after AAPH treatment but not in the LKO cells (Figure 2B,C). Representative traces (Figure 2D) from the Seahorse Analyzer showed that AAPH-induced oxidative stress decreased the OCR in response to the mitochondrial stress test (MST) paradigm in the WT cells but not in the LKO cells. The summary data from the repeated experiments confirm that basal respiration (Figure 2E), maximal respiration (Figure 2F), and ATP production (Figure 2G) were all significantly decreased in the WT cells after oxidative stress as compared to the LKO cells. We did not notice any change in proton leak between groups (Figure 2H).

### 3.3. LRP1 Deficiency Protects the Cell from Oxidative Stress and Mitochondrial Fragmentation

Oxidative stress is known to induce excessive mitochondrial fission [33,34]. To test whether LRP1 deficiency protects mitostasis after oxidative stress, we induced oxidative stress in the WT and LKO cells using AAPH and then measured the mitochondrial network via staining with MitoTracker Green. As expected, oxidative stress increased mitochondrial fragmentation in the WT cells (Figure 3A), as observed in the ipsilateral injured mtD2g brains (Figure 1A,B). Interestingly, the LKO cells maintained their mitochondrial length and network significantly better than the WT cells (Figure 3A). We quantified the mean mitochondrial length and branch network using the MiNA macro in ImageJ. Both parameters were significantly decreased in the WT cells after AAPH treatment but not in the LKO cells that were protected from such changes during oxidative stress (Figure 3B). Cellular antioxidants neutralize oxidative stress; however, excessive oxidative stress can still accumulate and damage cellular components. To evaluate whether LKO cells have a better ROS buffering capacity, we stained the cells with photosensitive DCHF-DA dye and measured the cellular ROS levels using a fluorescent microscope (Figure 3C). Again, AAPH treatment significantly increased the ROS levels in the WT cells compared to the LKO cells (Figure 3D). LRP1 responds to physiological insults such as hypoxia and injury that increase its expression level [24,25,26]. Our results suggest that oxidative stress also significantly increases LRP1 expression in WT cells (Figure 3E,F).

### 3.4. LRP1 Deficiency Protects Mitochondrial Complex Integrity, DNA Copy Number, and Cell Growth following Oxidative Stress

ROS can damage mitochondrial electron transport chain (ETC) proteins and DNA (nuclear and mitochondrial) and impair mitochondrial function to synthesize ATP for regular cell metabolism [35]. To determine the levels of ETC protein expression, cell lysates from the WT and LKO cells were subjected to Western blot analysis using a cocktail antibody mix against mitochondrial complex proteins (NDUFB8-complex I; SDHB-complex II; UQCRC2-complex III; ATP5A-complex V) (Figure 4A). We noticed that complex I protein significantly decreased with oxidative stress in the WT cells compared to the LKO cells. Importantly, LRP1 deficiency itself increased complex I and II protein expression as compared to the WT cells (Figure 4B,C). We did not notice any significant change in complex III and V expression levels between experimental groups (Figure 4D,E). In addition, the expressions of NDUFS1, another nuclear-encoded complex I subunit, and PDGFR-β were also upregulated significantly by LRP1 deficiency (Figure 4F–H). However, oxidative stress did not decrease NDUFS1 expression, as in the case of NDUFB8. As ROS accumulates, nuclear DNA damage can impinge on cell growth, resulting in cell cycle arrest or apoptosis depending on the cell type and stage of the cell cycle in which the insult occurs [36]. When we analyzed cell proliferation, we found that WT cell growth was significantly decreased following treatment, whereas LKO cell proliferation was not affected by oxidative stress (Figure 4I,J). Free radicals can damage mtDNA and decrease the mtDNA copy number [37]. Consistent with previous reports, oxidative stress significantly decreased the mtDNA copy number in the WT cells following oxidative stress, but interestingly, LRP1 deficiency blocked the phenotype (Figure 4K). 

### 3.5. LRP1 Deficiency Upregulates Genes Related to Mitochondrial Biogenesis and Antioxidants

Taken together, our results suggest that, following oxidative stress, the LKO cells were protected against mitochondrial damage and cell growth arrest, as compared to the WT cells. Hence, in the next set of experiments, we assessed the expression of genes involved in mitochondrial biogenesis and antioxidants at various mRNA levels (Table 1) using Taqman probe-based qRT-PCR. Our results indicate that compared to the WT cells, most genes involved in mitochondrial biogenesis with antioxidant properties were significantly upregulated in the LKO cells, especially the *TFAM*, *Nrf2*, *Pink1*, *ATP6*, *Dnm1*, *COX*, *Ucp2*, and *Sirt1* mRNA levels (Figure 5A,B). In the WT cells, AAPH treatment significantly increased the expression of *Nrf1* and *ND5/6* (Figure 5A,B). Moreover, at the protein level, the TFAM, PGC1α (Figure 5C,D), and PDGFR-β (Figure 4E,G) expressions were increased in the LKO cells compared to the WT cells according to Western blot analysis. Meanwhile, oxidative stress significantly decreased PGC1a expression but not TFAM expression (Figure 5C,D).

## 4. Discussion

Physiological insults such as TBI, ischemia-reperfusion, and stroke generate oxidative stress and mitochondrial dysfunction in the brain. Our research team is primarily focused on identifying targets for mitochondria-targeted therapeutics (mitoceuticals) to mitigate the pathological sequelae of TBI. In the past, we demonstrated that therapeutic interventions targeting mitochondrial-derived oxidative damage, such as antioxidants [38], mild uncouplers [39], and mitochondrial biogenesis activators [28], improve pathological outcomes in TBI models. Numerous studies demonstrate that mitochondria are dynamic cellular organelles that undergo continuous fusion and fission processes to meet energy demands, and any imbalance will be deleterious to cell survival [34,40,41]. Mitochondria can change their morphology very rapidly in response to any stress stimulus so as to maintain mitostasis [33]. Thus far, mitochondrial dynamics have been studied using different methods ranging from gene and protein expressions to proteomics after TBI [42,43,44]. However, only cryo-electron microscopy has been used as an imaging technique to visualize mitochondrial dynamics in processed tissue sections. In this study, we used a novel technique to measure mitochondrial morphometric dynamics in mitochondrial reporter mtD2g mouse brains after TBI. Furthermore, using Imaris and ImageJ, we resolved the mitochondrial structure in the mouse brain using confocal microscopy by measuring the size, shape, and, to some extent, the network of the mitochondria. Increased spherical-shaped and fragmented mitochondria were found around the ipsilateral cortex, while elongated rod-like mitochondria were observed in the corresponding contralateral cortex. In our in vitro model, very similar fragmented, spherical-shaped, shorter, and less complex mitochondrial networks were seen in the WT cells after oxidative stress. These results are in accordance with the known mitochondrial phenotypes following oxidative stress and brain injury.

LRP1 is highly expressed in the central and peripheral nervous systems, and it is differentially expressed in both neuronal and glial cell populations in developing and adult mouse brains [45]. The function of LRP1 in neurons has been studied using neuron- specific Cre, Synapsin I Cre (SynCre), and CamK2 Cre mice [46,47]. Synapsin I promoter is active in differentiated neurons and starts to express Cre recombinase from embryonic day 12.5 [48]. SynCre/LRP1^f/f^ mice appear to be normal without any defects in the nervous system, suggesting that LRP1 is not critical in differentiated neurons. However, SynCre/LRP1^f/f^ mice experienced tremors and dystonia due to a deficit in neurotransmission at around 6 months of age [46]. In CamK2Cre/LRP1^f/f^ mice, LRP1 deletion starts only between 3 and 6 months of age, and few behavioral abnormalities have been reported as compared to SynCre/LRP1^f/f^ mice. However, aged CamK2cre/LRP1^f/f^ mice showed some abnormalities in lipid metabolism and neuronal spine density at around 12–18 months of age, suggesting that the deletion of LRP1 at adult age is not critical for neurotransmission [47]. The majority of research focused on LRP1 function has been performed with the aim of understanding amyloid-beta (Aβ) trafficking in the blood–brain barrier (BBB), as LRP1 plays a predominant role in both Aβ production and clearance in Alzheimer’s disease (AD). However, the overall Aβ pathology is reduced in LRP1 knockout mice brains [49]. The overexpression of the LRP1 mini-receptor increases Aβ42 accumulation and decreases neuronal viability [50], and LRP1 deletion in neurons decreases α-synuclein and Aβ uptake and spread [15,50]. In microglia, LRP1 modulates inflammation through sLRP1 and LRP1-ICD [24,51]. It is believed that sLRP1 is robustly proinflammatory in nature and presents at an increased concentration in the CSF of aged persons [52] and plasma of rheumatoid arthritis patients [53]. The deletion of LRP1 in microglia decreases sLRP1 and blocks neuro-inflammation in the peripheral nerve injury model [10]. Taken together, these studies suggest that in the adult brain, the deletion of LRP1 in neurons or glial cells could be beneficial under certain circumstances. In addition to other proposed mechanisms proposed, LRP1 could have detrimental effects on brain metabolism after an insult, and mitochondrial-associated improvement following LRP1 deletion could, in part, explain the process of disease mitigation.

LRP1 is a negative regulator of the differentiation of OPCs in the adult mouse brain, and deleting the LRP1 gene increases the number of newborn mature oligodendrocytes [17]. In fibroblasts and smooth muscle cells, the deletion of LRP1 activates TGF-β and PDGFR-β signaling [14], and TGF-β signaling regulates the initiation of oligodendrocyte myelination processes [54]. Interestingly, TGF-β signaling itself upregulates the mitochondrial mass through peroxisome proliferator-activated receptor-gamma coactivator-1 alpha (PGC-1α) [55] and stimulates mitochondrial oxidative phosphorylation [56]. We reported that the β_2_-adrenoceptor agonist, Formoterol, mitigates TBI and spinal cord injury outcomes by improving mitochondrial biogenesis and mtDNA through PGC-1α [28,57]. Additionally, transcription factor A (TFAM) overexpression inhibits mitochondrial oxidative stress and improves mitochondrial function in the brain and heart [58,59]. Our results show that LRP1 deficiency increased PDGFR-β, PGC-1α, and TFAM expression at the protein level, suggesting that LRP1 may negatively regulate mitochondrial biogenesis. At the mRNA level, although LRP1 deficiency upregulated genes related to antioxidants and mitochondrial biogenesis (*TFAM*, *Nrf2*, *Pink1*, *ATP6*, *Dnm1*, *COX*, *Ucp2*, and *Sirt1*), we did not see any change in mitochondrial respiration without oxidative stress, suggesting that without oxidative stress, the cells did not respire at their maximum capacity. This mitochondrial phenomenon has previously been reported in macrophages, where exogenous mitochondrial transfer to healthy cells did not increase aerobic respiration, but metabolically stressed cells utilized exogenously transferred mitochondria to restore aerobic respiration [60]. We reported similar observations after TBI, where changes in state III and state V respiration were observed in post-injury metabolic crisis [28,61,62].

LRP1 is essential, and its deletion is embryonically lethal [63]. In this study, we specifically explored the role of LRP1 in normal mitochondrial function under oxidative stress conditions in vitro. Oxidative stress damages ETC complex proteins [35]. In TBI and ischemia-reperfusion, complex I activity is decreased [64,65]. In our study, we found that LRP1 deficiency increases ETC complex 1 and II expression and that WT cells are more vulnerable to oxidative damage than LKO cells. We confirmed this phenotype by visualizing the mitochondrial structure with a confocal microscope, quantifying DNA copy numbers via PCR, delineating mitochondrial function using a Seahorse assay, and analyzing complex protein expression via Western blot. Nevertheless, although this study strongly supports the hypothesis that LRP1 deficiency protects cells against oxidative stress by improving mitochondrial function, the mechanisms remain unclear.

We speculate that LRP1 deficiency removes negative feedback on TGF-β and also upregulates NRF-2/PGC-1α signaling, which increases mitochondrial biogenesis and antioxidants. LRP1 is mainly involved in lipid and cholesterol metabolism. We did see increased maximal respiration when we used palmitic acid and glucose as substrates (Appendix A). Deleting LRP1 could also change the lipoprotein and cholesterol composition of the plasma and mitochondrial membranes. Around 25% of the inner mitochondrial membrane is made up of a phospholipid called cardiolipin, and the oxidization of cardiolipin can significantly change mitochondrial function in the brain [66]. We suspect that the change in phospholipid composition may increase resistance to lipid peroxidation, which prevents mitochondrial swelling and damage. LRP1-ICD is known to regulate the transcription of inflammatory genes in the nucleus [67]. It is possible that deleting LRP1 may also alter the expression of genes involved in mitochondrial biogenesis. In our study, we found that the prophylactic overexpression of genes related to mitochondrial biogenesis in LKO cells may, in part, afford some protection against oxidative stress. One limitation of this study is that most of the experiments were conducted using fibroblast cells. However, the existing evidence from mice studies supports our theory that LRP1 deficiency could be beneficial in mitigating the pathophysiology of TBI. Further studies on neuron/glial cell-specific LRP-deficient mice are required to elucidate the mechanism of LRP1-mediated benefits in TBI and neurodegenerative disease pathological outcomes.

## 5. Conclusions

In conclusion, our results demonstrate that LRP1 deficiency preserves mitostasis and mitochondrial function following oxidative stress by increasing mitochondrial biogenesis and antioxidant gene expression. Hence, targeting LRP1 could be a potential strategy against oxidative stress in TBI and possibly other neurodegenerative diseases. Importantly, the mtD2g transgenic mouse model is a useful tool for studying mitostasis using confocal microscopy following TBI in concert with other therapeutic interventions.

## Figures and Tables

**Figure 1 cells-12-01445-f001:**
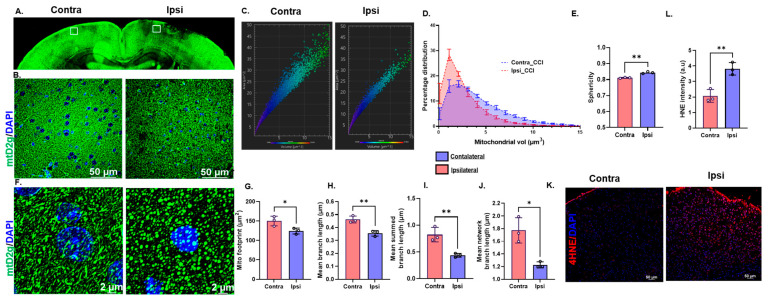
Traumatic brain injury induces oxidative stress and mitochondrial fragmentation. (**A**,**B**), Representative wide-field fluorescence images (20×; stitched large image) and confocal (60× oil) micrographs of mtD2g mice showing contra- and ipsi-cortex regions 24 h post-TBI. (**C**), Representative volume–area distribution graph of individual mitochondria from contra- and ipsi-cortex regions of the mtD2g mice brain 24 h post-TBI. (**D**,**E**), Graphical representation of the percent mitochondrial volume distribution (**D**) and sphericity (**E**) of the mitochondria (**F**). Representative confocal images used for mitochondrial network analysis (MiNA); green—mtD2g, and blue—DAPI. (**G**–**J**), Analysis of the mitochondrial network (mito-footprint, mean branch length, mean summed branch length, and mean network branch length) using the MiNA macro in ImageJ. (**K**), Representative immunofluorescence micrographs of 4-hydroxynonenal (4HNE) staining (red—4HNE, and blue—DAPI). (**L**), Quantification of the 4HNE fluorescence intensity of brain sections using ImageJ. All the experiments were performed using brain sections from mtD2 mice 24 h post-TBI (*n* = 3/group; open circules (o) represent individual data points). Data are represented as mean ± SD. *p* ≤ 0.05 *; *p* ≤ 0.01 ** using the unpaired *t*-test (**D**,**E**,**G**–**K**,**L**).

**Figure 2 cells-12-01445-f002:**
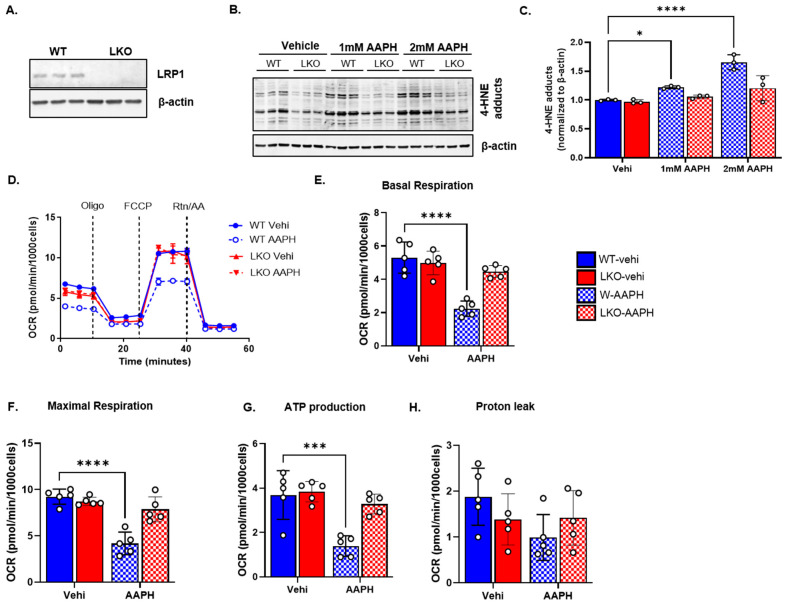
LRP1 deficiency improves mitochondrial bioenergetics following oxidative stress. (**A**), Representative LRP1 Western blot from WT and LKO cells showing complete knockout of LRP1. (**B**,**C**), Representative Western blot and quantification of 4HNE adducts from WT and LKO cells (*n* = 3 independent preparations/group; normalized to WT) treated with vehicle and AAPH. (**D**), Representative traces of mito-stress test using different respiratory chain inhibitors (oligomycin and Rotenone + Antimycin A) and uncoupler (FCCP) from WT and LKO cells treated with vehicle and AAPH. (**E**–**H**), Summary of basal respiration, maximal respiration, ATP production, and proton leak obtained using Seahorse XFe96, respectively (*n* = 5 independent preparations/group; open circles (o) represent individual data points). Data are represented as mean ± SEM. *p* ≤ 0.05 *; *p* ≤ 0.001 ***; *p* ≤ 0.0001 **** using two-way ANOVA with Sidak’s post hoc (**C**,**E**–**H**).

**Figure 3 cells-12-01445-f003:**
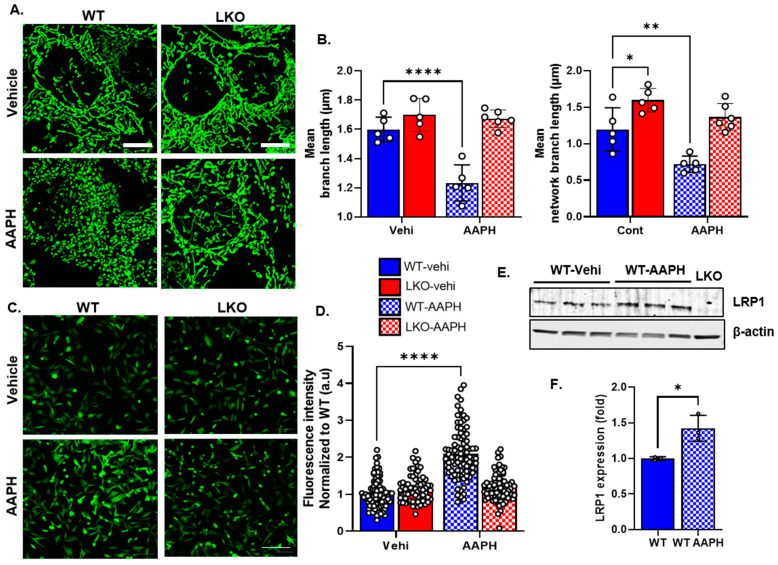
LRP1 deficiency protects the cell from oxidative-stress-induced mitochondrial fragmentation. (**A**), Representative MitoTracker-Green-stained micrographs of WT and LKO cells treated with vehicle and AAPH. (**B**). Bar graph representing the mitochondrial morphology (mean branch length and mean network branch length) analyzed based on confocal images using the MiNA macro in ImageJ (*n* = 5–6 random microscopic fields/group; 50–60 cells/group). (**C**), Representative photosensitive DCHF-DA stained micrographs of WT and LKO cells treated with vehicle and AAPH. (**D**), Bar graph summarizing average ROS levels (*n* = 80–100 cells from 8–10 microscopic fields/group); normalized to WT. (**E**,**F**), Representative LRP1 Western blot and quantification of WT and WT cells treated with AAPH (*n* = 3/group; normalized to WT vehicle). LKO cells were used as a negative control. Open circles (o) represent individual data points. Data are represented as mean ± SEM. *p* ≤ 0.05 *; *p* ≤ 0.01 **; *p* ≤ 0.0001 **** using two-way ANOVA with Sidak’s post hoc (**B**,**D**); unpaired *t*-test (**F**).

**Figure 4 cells-12-01445-f004:**
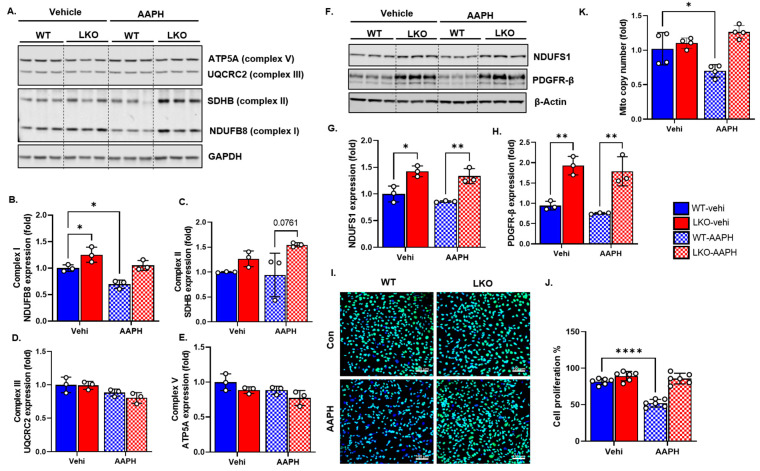
LRP1 deficiency protects mitochondrial complex integrity, DNA copy number, and cell growth following oxidative stress. (**A**), Representative Western blot micrograph of WT and LKO cells treated with vehicle and AAPH (*n* = 3 independent preparations/group) for different mitochondrial respiratory chain complex proteins. (**B**–**E**), Quantification of complex I (NDUFSB8), complex II (SDHB), complex III (UQCRC2), and complex V (ATP5A) normalized to WT. (**F**–**H**), Representative Western blot micrograph and quantification of WT and LKO cells (*n* = 3 independent preparations/group; normalized to WT) treated with vehicle and AAPH for NDUFS1 (complex I) protein and PDGFR-β. (**I**,**J**), Representative micrographs from the Click-iT EdU assay for cell proliferation. Green color represents EdU incorporation and blue represents the DAPI counterstain for the nucleus. Summary represented as percent cell proliferation normalized to the total nucleus count in each group (*n* = 6 random microscopic fields/group; each field had over 200 cells). (**K**), Fold change in the mitochondrial copy numbers of WT and LKO cells treated with vehicle and AAPH (2^−ΔΔCT^ method; normalized to WT). Open circles (o) represent individual data points. Data are represented as mean ± SEM. *p* ≤ 0.05 *; *p* ≤ 0.01 **; *p* ≤ 0.0001 **** using two-way ANOVA with Sidak’s post hoc test.

**Figure 5 cells-12-01445-f005:**
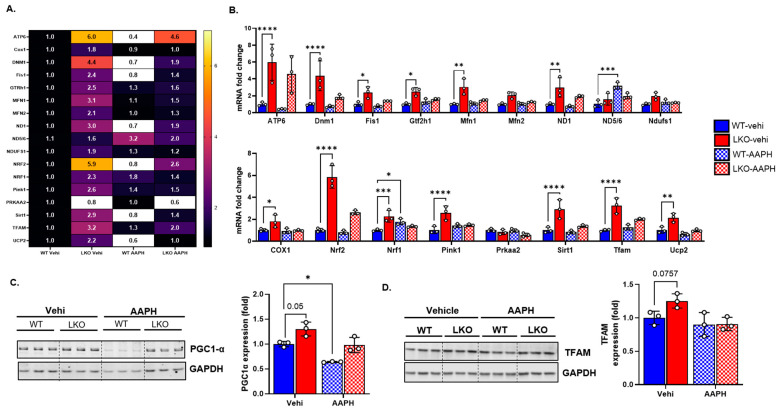
LRP1 deficiency increases mitochondrial biogenesis and antioxidant genes. (**A**,**B**), Heat map and bar graph representing mRNA fold change in mitochondrial biogenesis- and antioxidant-related genes from WT and LKO cells treated with vehicle and AAPH (2^−ΔΔCT^ method; normalized to WT vehicle), with 18s RNA used as an endogenous control (*n* = 3 independent preparations/group). (**C**), Representative Western blot micrograph and quantification of WT and LKO cells treated with vehicle and AAPH for PGC1α. (**D**), Representative Western blot micrograph and ImageJ quantification of the mean intensity of WT and LKO cells under normal and oxidative conditions for TFAM (*n* = 3 independent preparations/group). Data are represented as mean ± SD. *p* ≤ 0.05 *; *p* ≤ 0.01 **; *p* ≤ 0.001 ***; *p* ≤ 0.0001 **** using two-way ANOVA with Sidak’s post hoc test (**B**–**D**).

**Table 1 cells-12-01445-t001:** List of genes and assay IDs used for mRNA analysis.

S. No.	Gene Symbol		Assay ID
1	TFAM	Transcription Factor A, Mitochondrial	Mm00447485_m1
2	Nrf1	Nuclear respiratory factor 1	Mm01135606_m1
3	Prkaa2	Protein Kinase AMP-Activated Catalytic Subunit Alpha 2	Mm01264789_m1
4	ND1	NADH-ubiquinone oxidoreductase chain 1	Mm04225274_s1
5	Ucp2	Uncoupling protein 2	Mm00627599_m1
6	Pink1	PTEN-induced putative kinase 1	Mm00550827
7	Gtf2h1	General transcription factor IIH subunit 1	Mm00500417_m1
8	Mfn1	Mitofusin 1	Mm00612599_m1
9	Fis1	Fission protein 1	Mm00481580
10	ATP6	ATP Synthase Membrane Subunit 6	Mm03649417_g1
11	Nfe212 (Nrf2)	Nuclear factor erythroid 2-related factor 2	Mm00477784_m1
12	Sirt1	Sirtuin 1	Mm01168521_m1
13	Mfn2	Mitofusin 2	Mm00500120
14	Dnm1	Dynamin 1	Mm01342903_m1
15	COX1	Cytochrome c oxidase I	Mm04225243_g1
16	Ndufs1	NADH:Ubiquinone Oxidoreductase Core Subunit S1	Mm005236040_m1
17	ND6/ND5	NADH-ubiquinone oxidoreductase chain 6 protein/5	Mm04225325_g1
18	18sRNA		Mm03928990_g1

## Data Availability

The original contributions presented in the study are included in the article/Appendix A. Further inquiries can be directed to the corresponding author.

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
