# Peer review of "LRP1 Deficiency Promotes Mitostasis in Response to Oxidative Stress: Implications for Mitochondrial Targeting after Traumatic Brain Injury"

_cells, 2023, doi:10.3390/cells12101445_

Round 1

Reviewer 1 Report

In the manuscript entitled “LRP1 deficiency promotes mitostasis in response to oxidative stress: Implications for mitochondrial targeting after traumatic brain injury” by Velmurugan G.P. et al., Authors describe the role of LRP-1 in mitochondrial dysfunction induced by TBI. Using in vivo and in vitro experiments, results suggest that LRP-1 inhibition is beneficial for the post-injury brain and that it may represent a new target for potential pharmacological treatments.

Although the manuscript reports data of interest in the field of TBI, there are several concerns that Authors should carefully address before manuscript is acceptable for the publication, including a careful reading of the text to eliminate typewriting errors and to allow matching of the text with figures.

Abstract

The Abstract should be rewritten, following the Instruction for the Authors of the Journal reported below, “The abstract should be a total of about 200 words maximum. The abstract should be a single paragraph and should follow the style of structured abstracts, but without headings: 1) Background: Place the question addressed in a broad context and highlight the purpose of the study; 2) Methods: Describe briefly the main methods or treatments applied. Include any relevant preregistration numbers, and species and strains of any animals used; 3) Results: Summarize the article's main findings; and 4) Conclusion: Indicate the main conclusions or interpretations. The abstract should be an objective representation of the article: it must not contain results which are not presented and substantiated in the main text and should not exaggerate the main conclusions.”

Introduction

Lines 44-46, please add some references in support to these notions.

Lines 46-48, Authors affirm that “Though there are studies focused on understanding the mitochondrial dysfunction in TBI and other neurodegenerative diseases, paucity of relevant information on the specific impact of oxidative damage warrants further studies”. This sentence should be re-written. In PubMed, using the terms traumatic brain injury and oxidative stress, it is shown that 1367 match these words. That is, we have abundant information on the mechanisms triggering oxidative stress and the molecular targets of ROS, even in TBI. What it is missing is a really effective pharmacological treatment for TBI.

Line 57, delete “is”.

Line 63, add “is” after “deletion”.

Results

The result section should be dedicated to comment the experimental data. In this light, my suggestion is to delete “In the process of ATP production, mitochondria generate reactive oxygen species (ROS) as a byproduct. Over production of ROS generates reactive hydroxyl radicals that react with unsaturated fatty acids in membrane proteins causing lipid peroxidation and mitochondrial dysfunction [28]. Structural and functional mitochondrial damages links to increased oxidative stress in neurons in the acute phase of TBI [29]. Oxidative stress increases mitochondrial fission and excessive fission causes metabolic deficits and neuronal death following TBI [29].” Lines 222-227

The same can be applied for line 263 “LRP1 plays a critical role in the developing and adult brain [30]. However, the role of LRP1 in oxidative stress and mitochondrial function is not fully understood.” That should therefore be deleted.

From line 269 to line 273, there are comments to metabolic data and Figure 1 (1C, 1D, 1E) is mentioned throughout these lines. However, metabolic data are in different panels of Figure 2. Great confusion has been done by the Authors, who are cordially invited to make all the corrections needed to match text with Figures!! Delete lines 320-322, up to ref [33].

In all the figure, results are expressed as means ± SEM. In order to allow readers to better appreciate data dispersion, figures should be modified in order to present data as means ± SD.

Discussion

In lines 390-391, Authors affirm that “So far, mitochondrial dynamics have only been studied using cryo- electron microscopy in processed tissue sections.” This is not true. Here some references that Authors should

check (and probably cite) after having modified the text, in which mitochondrial dynamics has been studied using different methods, from gene and protein expressions to proteomic:

Niu F, Dong J, Xu X, Zhang B, Liu B. Mitochondrial Division Inhibitor 1 Prevents Early-Stage Induction of Mitophagy and Accelerated Cell Death in a Rat Model of Moderate Controlled Cortical Impact Brain Injury. World Neurosurg. 2019 Feb;122:e1090-e1101. doi: 10.1016/j.wneu.2018.10.236.

Cui W, Wu X, Shi Y, Guo W, Luo J, Liu H, Zheng L, Du Y, Wang P, Wang Q, Feng D, Ge S, Qu Y. 20-HETE synthesis inhibition attenuates traumatic brain injury-induced mitochondrial dysfunction and neuronal apoptosis via the SIRT1/PGC-1α pathway: A translational study. Cell Prolif. 2021 Feb;54(2):e12964. doi: 10.1111/cpr.12964.

Song H, Chen M, Chen C, Cui J, Johnson CE, Cheng J, Wang X, Swerdlow RH, DePalma RG, Xia W, Gu Z. Proteomic Analysis and Biochemical Correlates of Mitochondrial Dysfunction after Low-Intensity Primary Blast Exposure. J Neurotrauma. 2019 May 15;36(10):1591-1605. doi: 10.1089/neu.2018.6114.

Wu Q, Gao C, Wang H, Zhang X, Li Q, Gu Z, Shi X, Cui Y, Wang T, Chen X, Wang X, Luo C, Tao L. Mdivi-1 alleviates blood-brain barrier disruption and cell death in experimental traumatic brain injury by mitigating autophagy dysfunction and mitophagy activation. Int J Biochem Cell Biol. 2018 Jan;94:44-55. doi: 10.1016/j.biocel.2017.11.007.

Di Pietro V, Lazzarino G, Amorini AM, Signoretti S, Hill LJ, Porto E, Tavazzi B, Lazzarino G, Belli A. Fusion or Fission: The Destiny of Mitochondria In Traumatic Brain Injury of Different Severities. Sci Rep. 2017 Aug 23;7(1):9189. doi: 10.1038/s41598-017-09587-2.

In the manuscript entitled “LRP1 deficiency promotes mitostasis in response to oxidative stress: Implications for mitochondrial targeting after traumatic brain injury” by Velmurugan G.P. et al., Authors describe the role of LRP-1 in mitochondrial dysfunction induced by TBI. Using in vivo and in vitro experiments, results suggest that LRP-1 inhibition is beneficial for the post-injury brain and that it may represent a new target for potential pharmacological treatments.

Although the manuscript reports data of interest in the field of TBI, there are several concerns that Authors should carefully address before manuscript is acceptable for the publication, including a careful reading of the text to eliminate typewriting errors and to allow matching of the text with figures.

Abstract

The Abstract should be rewritten, following the Instruction for the Authors of the Journal reported below, “The abstract should be a total of about 200 words maximum. The abstract should be a single paragraph and should follow the style of structured abstracts, but without headings: 1) Background: Place the question addressed in a broad context and highlight the purpose of the study; 2) Methods: Describe briefly the main methods or treatments applied. Include any relevant preregistration numbers, and species and strains of any animals used; 3) Results: Summarize the article's main findings; and 4) Conclusion: Indicate the main conclusions or interpretations. The abstract should be an objective representation of the article: it must not contain results which are not presented and substantiated in the main text and should not exaggerate the main conclusions.”

Introduction

Lines 44-46, please add some references in support to these notions.

Lines 46-48, Authors affirm that “Though there are studies focused on understanding the mitochondrial dysfunction in TBI and other neurodegenerative diseases, paucity of relevant information on the specific impact of oxidative damage warrants further studies”. This sentence should be re-written. In PubMed, using the terms traumatic brain injury and oxidative stress, it is shown that 1367 match these words. That is, we have abundant information on the mechanisms triggering oxidative stress and the molecular targets of ROS, even in TBI. What it is missing is a really effective pharmacological treatment for TBI.

Line 57, delete “is”.

Line 63, add “is” after “deletion”.

Results

The result section should be dedicated to comment the experimental data. In this light, my suggestion is to delete “In the process of ATP production, mitochondria generate reactive oxygen species (ROS) as a byproduct. Over production of ROS generates reactive hydroxyl radicals that react with unsaturated fatty acids in membrane proteins causing lipid peroxidation and mitochondrial dysfunction [28]. Structural and functional mitochondrial damages links to increased oxidative stress in neurons in the acute phase of TBI [29]. Oxidative stress increases mitochondrial fission and excessive fission causes metabolic deficits and neuronal death following TBI [29].” Lines 222-227

The same can be applied for line 263 “LRP1 plays a critical role in the developing and adult brain [30]. However, the role of LRP1 in oxidative stress and mitochondrial function is not fully understood.” That should therefore be deleted.

From line 269 to line 273, there are comments to metabolic data and Figure 1 (1C, 1D, 1E) is mentioned throughout these lines. However, metabolic data are in different panels of Figure 2. Great confusion has been done by the Authors, who are cordially invited to make all the corrections needed to match text with Figures!! Delete lines 320-322, up to ref [33].

In all the figure, results are expressed as means ± SEM. In order to allow readers to better appreciate data dispersion, figures should be modified in order to present data as means ± SD.

Discussion

In lines 390-391, Authors affirm that “So far, mitochondrial dynamics have only been studied using cryo- electron microscopy in processed tissue sections.” This is not true. Here some references that Authors should

check (and probably cite) after having modified the text, in which mitochondrial dynamics has been studied using different methods, from gene and protein expressions to proteomic:

Niu F, Dong J, Xu X, Zhang B, Liu B. Mitochondrial Division Inhibitor 1 Prevents Early-Stage Induction of Mitophagy and Accelerated Cell Death in a Rat Model of Moderate Controlled Cortical Impact Brain Injury. World Neurosurg. 2019 Feb;122:e1090-e1101. doi: 10.1016/j.wneu.2018.10.236.

Cui W, Wu X, Shi Y, Guo W, Luo J, Liu H, Zheng L, Du Y, Wang P, Wang Q, Feng D, Ge S, Qu Y. 20-HETE synthesis inhibition attenuates traumatic brain injury-induced mitochondrial dysfunction and neuronal apoptosis via the SIRT1/PGC-1α pathway: A translational study. Cell Prolif. 2021 Feb;54(2):e12964. doi: 10.1111/cpr.12964.

Song H, Chen M, Chen C, Cui J, Johnson CE, Cheng J, Wang X, Swerdlow RH, DePalma RG, Xia W, Gu Z. Proteomic Analysis and Biochemical Correlates of Mitochondrial Dysfunction after Low-Intensity Primary Blast Exposure. J Neurotrauma. 2019 May 15;36(10):1591-1605. doi: 10.1089/neu.2018.6114.

Wu Q, Gao C, Wang H, Zhang X, Li Q, Gu Z, Shi X, Cui Y, Wang T, Chen X, Wang X, Luo C, Tao L. Mdivi-1 alleviates blood-brain barrier disruption and cell death in experimental traumatic brain injury by mitigating autophagy dysfunction and mitophagy activation. Int J Biochem Cell Biol. 2018 Jan;94:44-55. doi: 10.1016/j.biocel.2017.11.007.

Di Pietro V, Lazzarino G, Amorini AM, Signoretti S, Hill LJ, Porto E, Tavazzi B, Lazzarino G, Belli A. Fusion or Fission: The Destiny of Mitochondria In Traumatic Brain Injury of Different Severities. Sci Rep. 2017 Aug 23;7(1):9189. doi: 10.1038/s41598-017-09587-2.

Author Response

Response to Reviewer 1 Comments

The Abstract should be rewritten, following the Instruction for the Authors of the Journal reported below, “The abstract should be a total of about 200 words maximum. The abstract should be a single paragraph and should follow the style of structured abstracts, but without headings: 1) Background: Place the question addressed in a broad context and highlight the purpose of the study; 2) Methods: Describe briefly the main methods or treatments applied. Include any relevant preregistration numbers, and species and strains of any animals used; 3) Results: Summarize the article's main findings; and 4) Conclusion: Indicate the main conclusions or interpretations. The abstract should be an objective representation of the article: it must not contain results which are not presented and substantiated in the main text and should not exaggerate the main conclusions.”

Response 1: Thanks for the comment the abstract has been changed completely to 200 words maximum as per journal guidelines.

Introduction

Lines 44-46, please add some references in support to these notions.

Reference added as the reviewer suggested

Lines 46-48, Authors affirm that “Though there are studies focused on understanding the mitochondrial dysfunction in TBI and other neurodegenerative diseases, paucity of relevant information on the specific impact of oxidative damage warrants further studies”. This sentence should be re-written. In PubMed, using the terms traumatic brain injury and oxidative stress, it is shown that 1367 match these words. That is, we have abundant information on the mechanisms triggering oxidative stress and the molecular targets of ROS, even in TBI. What it is missing is a really effective pharmacological treatment for TBI.

The sentence as has been rewritten as the reviewer suggested

Line 57, delete “is”.

Corrected

Line 63, add “is” after “deletion”.

Corrected

Results

The result section should be dedicated to comment the experimental data. In this light, my suggestion is to delete “In the process of ATP production, mitochondria generate reactive oxygen species (ROS) as a byproduct. Over production of ROS generates reactive hydroxyl radicals that react with unsaturated fatty acids in membrane proteins causing lipid peroxidation and mitochondrial dysfunction [28]. Structural and functional mitochondrial damages links to increased oxidative stress in neurons in the acute phase of TBI [29]. Oxidative stress increases mitochondrial fission and excessive fission causes metabolic deficits and neuronal death following TBI [29].” Lines 222-227

Corrected

The same can be applied for line 263 “LRP1 plays a critical role in the developing and adult brain [30]. However, the role of LRP1 in oxidative stress and mitochondrial function is not fully understood.” That should therefore be deleted.

Corrected

From line 269 to line 273, there are comments to metabolic data and Figure 1 (1C, 1D, 1E) is mentioned throughout these lines. However, metabolic data are in different panels of Figure 2. Great confusion has been done by the Authors, who are cordially invited to make all the corrections needed to match text with Figures!! Delete lines 320-322, up to ref [33].

Apologies for the confusion caused. Last-minute changes caused this and we have now edited figures, legends, and results for this figure.

In all the figure, results are expressed as means ± SEM. In order to allow readers to better appreciate data dispersion, figures should be modified in order to present data as means ± SD.

We changed all the graphs with Mean ± SD instead of Mean± SME and we have all the data points in the figures.

Discussion

In lines 390-391, Authors affirm that “So far, mitochondrial dynamics have only been studied using cryo- electron microscopy in processed tissue sections.” This is not true. Here some references that Authors should check (and probably cite) after having modified the text, in which mitochondrial dynamics has been studied using different methods, from gene and protein expressions to proteomic:

Thanks for these suggestions. The message was not correctly conveyed in the discussion. As the reviewer suggested we have included references and rewritten the sentence.

In the manuscript entitled “LRP1 deficiency promotes mitostasis in response to oxidative stress: Implications for mitochondrial targeting after traumatic brain injury” by Velmurugan G.P. et al., Authors describe the role of LRP-1 in mitochondrial dysfunction induced by TBI. Using in vivo and in vitro experiments, results suggest that LRP-1 inhibition is beneficial for the post-injury brain and that it may represent a new target for potential pharmacological treatments. Although the manuscript reports data of interest in the field of TBI, there are several concerns that Authors should carefully address before manuscript is acceptable for the publication, including a careful reading of the text to eliminate typewriting errors and to allow matching of the text with figures.

Thanks for all your valuable comments.  As the reviewer suggested we have carefully read the manuscript and corrected all the typographical errors. Figures, legends, and results are now matched correctly.

Reviewer 2 Report

In this article, Velmurugan et al. demonstrate that LRP1 deficiency in fibroblasts and knockout mice preserves mitostasis and mitochondrial function after oxidative stress by increasing mitochondrial biogenesis and the expression of genes related to the antioxidant response in a model of traumatic brain injury. The introduction and the study's objective are concise and clear, the methodology is explained correctly, and the conclusion is based on the results. Therefore it is suggested to accept the article after the following minor changes:

1.    Abstract: Define: LDL, TGF-β, PGC-1α, WT, LRP1, TFAM, PDGFR-β Nrf2, pink1, sirt1

2.    Line 40: Define FDA

3.    Methods: Indicate the weight of the animals, define: EdU, and indicate how much protein volume was loaded in the WB

4.    Figure 1: Review the figure caption; it is mentioned that D and E are histograms; however, D does not correspond to a histogram.

5.    There are inconsistencies with what is described in the text and images:

-       Line 267: it is mentioned that Figure 2A shows the representative traces of the Seahorse analyzer; however, figure 2A corresponds to a WB.

-       The panels of Figure 1 are cited in the description of the respirometry, but these correspond to Figure 2. Review carefully.

-       Cite in the text panels "A and G" of Figure 4

-       Figure 4 panel H: The scale bar is missing in each figure.

-       Figure 5: Indicate in the footer of the figure what the numbers used in the statistics of panels C and D mean.

6.    Panel F and G of Figure 4, what is the purpose of comparing the LKO-AAPH group vs. WT-vehi? It might appear that in the deficient LRP1 subjected to oxidative stress, NDUFS1 and PDGFR-β are increased; however, it is not so because the vehicle of that group (LKO-vehi) is similar to the LKO-AAPH group. If the objective is to show all the significant data that the statistical analyzes arrogate, it must be homogenized in all the results where there may be a difference between the LKO-AAPH and WT-vehi groups, for example, figure 5.

1.    Line 35: Change " to Center for Disease Control " to " to the Center for Disease Control". It seems that there is an article usage problem here.

2.    Line 63: Change " Though LRP1 plays critical role " to " Though LRP1 plays a critical role ". The noun phrase critical role seems to be missing a determiner before it. Consider adding an article.

3.    In all places where it is necessary: change “brain” by “the brain”. The noun phrase brain seems to be missing a determiner before it. Consider adding an article.

4. Line 152: Change " transferred to PVDF membrane" to " transferred to the PVDF membrane ".

5. Line 364-367: Change " Figure " to " Figures". It seems that Figure may not agree in number with other words in this phrase.

6.  Line 383: Change " pathological sequalae of TBI" to "pathological sequelae of TBI".

Author Response

Response to Reviewer 2 Comments

  1. Abstract: Define: LDL, TGF-β, PGC-1α, WT, LRP1, TFAM, PDGFR-β Nrf2, pink1, sirt1

Response 2: The abstract is completely rewritten and due to the word limit we excluded all the gene names. However, we added all the definitions in Table 1.

  1. Line 40: Define FDA

Definition added

  1. Methods: Indicate the weight of the animals, define: EdU, and indicate how much protein volume was loaded in the WB

As the reviewer suggested all the corrections were made in the main text.

  1. Figure 1: Review the figure caption; it is mentioned that D and E are histograms; however, D does not correspond to a histogram.

Apologies for the confusion caused. Last-minute changes caused this and we have now edited figures, legends, and results for this figure.

  1. There are inconsistencies with what is described in the text and images:

-       Line 267: it is mentioned that Figure 2A shows the representative traces of the Seahorse analyzer; however, figure 2A corresponds to a WB.

The same issue as above has been corrected

- The panels of Figure 1 are cited in the description of the respirometry, but these correspond to Figure 2. Review carefully.

The same issue as above has been corrected

-       Cite in the text panels "A and G" of Figure 4

Thanks for pointing out missing legends. In the revision Figure 4 A and G are cited.

 -       Figure 4 panel H: The scale bar is missing in each figure.

The scale bar was invisible due to the font size. Corrected and made it bigger for easy visibility.

-       Figure 5: Indicate in the footer of the figure what the numbers used in the statistics of panels C and D mean.

Added in the figure legends

  1. Panel F and G of Figure 4, what is the purpose of comparing the LKO-AAPH group vs. WT-vehi? It might appear that in the deficient LRP1 subjected to oxidative stress, NDUFS1 and PDGFR-β are increased; however, it is not so because the vehicle of that group (LKO-vehi) is similar to the LKO-AAPH group. If the objective is to show all the significant data that the statistical analyzes arrogate, it must be homogenized in all the results where there may be a difference between the LKO-AAPH and WT-vehi groups, for example, figure 5.

The graph changed to convey the results. We would like to show that oxidative stress (AAPH induced) is not changing NDUFS1, PDGFR-β expression in LKO but decreasing in WT cells. Though the expression is decreased in WT-AAPH compared to WT-Vehi, it is not significant. Since this panel describes mitochondrial complex proteins and cell growth we thought NDUFS1, PDGFR-β Western blot is more appropriate in this panel.

Comments on the Quality of English Language

  1. Line 35: Change " to Center for Disease Control " to " to the Center for Disease Control". It seems that there is an article usage problem here.

Corrected

  1. Line 63: Change " Though LRP1 plays critical role " to " Though LRP1 plays a critical role ". The noun phrase critical role seems to be missing a determiner before it. Consider adding an article.

Corrected

  1. In all places where it is necessary: change “brain” by “the brain”. The noun phrase brain seems to be missing a determiner before it. Consider adding an article.

Replaced wherever necessary.

  1. Line 152: Change " transferred to PVDF membrane" to " transferred to the PVDF membrane ".

Corrected

  1. Line 364-367: Change " Figure " to " Figures". It seems that Figure may not agree in number with other words in this phrase.

Corrected

  1. Line 383: Change " pathological sequalae of TBI" to "pathological sequelae of TBI".

Corrected